# Reliable Knowledge about Obesity Risk, Rather Than Personality, Is Associated with Positive Beliefs towards Obese People: Investigating Attitudes and Beliefs about Obesity, and Validating the Polish Versions of ATOP, BAOP and ORK–10 Scales

**DOI:** 10.3390/ijerph192214977

**Published:** 2022-11-14

**Authors:** Wojciech Styk, Ewa Wojtowicz, Szymon Zmorzynski

**Affiliations:** 1Department of Psychology, Medical University of Lublin, 20-059 Lublin, Poland; 2Department of Psychology, Institute of Pedagogy and Psychology, Warsaw Management University, 03-772 Warszawa, Poland; 3Department of Cancer Genetics with Cytogenetic Laboratory, Medical University of Lublin, 20-059 Lublin, Poland

**Keywords:** ATOP, BAOP, ORK–10, overweight, obesity, BMI, stereotypes, personality

## Abstract

Obesity has reached epidemic proportions. With the increase in the number of obese people, we have also witnessed a rise in the stigmatisation of this population. The aim of our study was to: (I) validate Polish versions of the attitude toward obese people (ATOP) scale, the beliefs about obese persons (BAOP) scale, and translate the obesity risk knowledge scale (ORK–10); (II) analyse the relationship between personality and the knowledge about obesity, as well as attitudes and beliefs towards obese people. Methods: The translation procedure was based on the principles of intercultural validation scales. The study was conducted on a group of 306 individuals, including 189 females and 117 males. Results: The original three-factor structure of the ATOP scale was confirmed in the Polish version. Factor analysis confirmed the one-factor structure of the BAOP scale in the Polish version. A very strong correlation was found between ATOP/BAOP and ORK–10. The correlation of personality with ATOP/BAOP scales was at a low level. Regression analysis indicated that knowledge of obesity risk predicted ATOP and BAOP by more than 58% and 50%, in turn, personality only 20% and 3.7%, respectively. Conclusion: The polish versions of ATOP, BAOP and ORK–10 scales are fully useful measurement tools. The knowledge about obesity risk is associated with beliefs and attitudes about obese people.

## 1. Introduction

In recent years, obesity has reached epidemic proportions contributing to the development of co-morbidities. Moreover, the psychosocial consequences of excess body weight have been observed. According to the World Health Organisation (WHO) (2021), the main cause of overweight and obesity is increased consumption of high-calorie foods with a concomitant decrease in physical activity such as inactive lifestyle, sedentary work, changes in the use of transport and progressive urbanisation. However, obesity is a complex, multifactorial disease in the development of which genetic and environmental factors cannot be ignored [1].

Various social beliefs about the causes of overweight and obesity have emerged with the increasing body mass of the population. The importance of these social beliefs for the wellbeing of overweight people is attributed to playing a key role in self-perception and social evaluations.

A number of studies indicated negative assessments of obese people, with women being more frequently affected than men [2,3,4]. Furthermore, negative assessments even occur in young children [5]. Obesity stigma and related discrimination have been documented in most key areas of people’s lives: work, health and physical activity; education; and interpersonal relationships [6,7,8,9,10]. The widespread stigmatisation of obese people is much greater than once thought. The view that obesity is a person’s choice and that weight loss is the responsibility of the individual has been a common socially held view [9]. This perceived responsibility concerns persistence in controlling food intake, weight and physical activity levels. This belief promotes stigma and consequently leads to discrimination in various societal areas. It can also limit the implementation of health-promoting changes by reducing the persistence and effectiveness of individuals in different situations [11,12]. In the case of people struggling with obesity, this can nullify efforts directed at body mass reduction and decrease the chances of harm limitation in their lives. For example, overweight women consumed three times more kilocalories after watching a stigmatising film than in the case of a neutral film [13,14]. It also appears that criticising overweight and obese young people and those at risk of becoming overweight was associated with higher weight and body fat gain over a 15-year period [14]. In this case, there is a “vicious cycle” in which stigmatisation has no positive effect and exacerbates the obesity problem even further [15].

Research conducted in Germany and the USA on social beliefs about the causes of obesity have shown that the cause of obesity can be culturally differentiated. Respondents in the USA were more strongly convinced of the socio-cultural causes of obesity, while in Germany, it was more often described as being due to over-eating and a lack of physical activity [16]. In contrast, people in the UK indicated that the causes of obesity were associated primarily with environmental factors and a lack of willpower rather than genetic factors [17]. The belief in the role of the state in preventing and treating obesity is linked to the attribution of the causes of overweight and obesity. Furthermore, a belief in the role of genes builds positive attitudes towards obesity treatments for free and educational campaigns. In contrast, positioning the cause of obesity in individual-dependent or environmental factor terms is associated with a stronger representation of stigmatising attitudes and greater support for healthy lifestyle campaigns, as well as attention to food composition and calorie content.

Negative attitudes among medical professionals are particularly important in the fight against obesity. In line with the idea of medical professions, they should encourage patients to control their body mass through proper education, assistance in achieving a health-promoting lifestyle and support in dealing with the consequences of too much body mass. However, research indicated that often students of medical sciences and medical professionals are characterised by negative beliefs regarding the motivation and discipline of patients [9,18,19,20,21,22]. As a consequence, patients are exposed to negative behaviour towards them, and the effectiveness of health interventions decreases significantly [23].

Consequently, people seeking solutions appropriate to their situation face problems related to not only their body mass but also difficulties arising from negative social attitudes towards them. The psychosocial determinants of overweight and obese people’s functioning are also reflected in contemporary research trends, social campaigns and practical actions. On the one hand, these tools focus on improving the wellbeing of individuals burdened by body mass stigma, and on the other hand, educate patients as well as the general public about the complex factors that lead to obesity.

The research on stereotypes and prejudice has resulted in a number of findings that support an understanding of the negative judgement phenomenon toward overweight and obese individuals. In the case of this population, the questions posed by researchers of these phenomena concerning their background and the practical possibilities of reducing the stereotypes’ impact on judgement formation are still relevant. While stereotypes and prejudices can be beneficial to certain groups due to their particular social harm, the analysis of their negative association with judgements plays a significant role in contemporary science.

Prejudice against obese people is defined as the tendency to construct judgements and evaluations on the basis of excessive body mass. Oppressive attitudes towards obese people appear to be more socially acceptable than other prejudices [24]. Quite often, excessive body mass is treated as a trait similar to other criteria on the basis of which negative social views are formed, such as race, disability or gender. However, the weight factor is not socially treated as stable and independent of the individual—people tend to discriminate against obese people because they believe that body mass is controllable and that obese people have a direct influence on their weight [5,8]. This can modify the processes of obesity judging—both in the case of concepts about oneself and the social perception of it.

Learning about personality causes can foster a better understanding of the stereotyping phenomenon and discrimination against obese people. One of the models more commonly used in research is the “Big Five” model. The study suggested that openness to experience and agreeableness were negatively correlated with biases [25]. Moreover, people with a high intensity of these traits were characterised by more positive attitudes to immigrants [26]. Particular importance is attributed to agreeableness as a trait that modifies prejudice and discriminatory behaviour [27,28].

In the case of environmental factors, the media (traditional and electronic) are crucial in formulating the views on obesity. Unsurprisingly, they can be a cause for the formation and spread of prejudices. They can, however, also be a valuable channel for their reduction and positive social change in terms of the perception of overweight people. The media are mainly accused of promoting ideal or unrealistic social norms regarding the body and criticising people who differ from this model [29]. Thus, negative judgements about overweight people are present in children and young people’s closest environment, especially family, peer groups and at school, as well as in all areas of adult functioning. The media, as mentioned previously, can also play a positive role in the obesity discourse by implementing educational activities aimed at changing attitudes towards obese people or increasing public awareness of the causes of excessive weight and health-promoting behaviours. Even a single exposure to an online educational module about obesity brings a small, short-term improvement in the attitudes of medical professionals [30]. In contrast, other studies also carried out among medical personnel did not show an association between knowledge, attitudes and prejudices toward obese people [31]. This opens up the question of whether a focus on education can be beneficial in the fight against obesity and its consequences and against discrimination against overweight people.

The aim of our study was to translate into Polish and validate scales to measure attitudes and prejudices towards obese people. Moreover, we validated the well-known obesity knowledge test. According to our knowledge, such tools are not available in Polish.

Furthermore, the aim of our research was also to check the factors co-occurring with obesity prejudice, taking into account the three-component structure of the ATOP scale. Previous studies and translations of the ATOP scale, despite the three-factor structure, were based only on the total score without analysing and exploring the structure of ATOP scale.

Our hypotheses concerned the relationship between personality and knowledge about obesity, as well as attitudes and prejudices towards obese people. Similar to the above studies on stereotypes, we hypothesised that openness and agreeableness would positively correlate with positive attitudes towards obese people [25,27,28]. Despite the indicated discrepancies in the research on the association between knowledge and prejudice, we also hypothesised that knowledge about obesity risk would be positively associated with favourable attitudes and lack of prejudice.

## 2. Materials and Methods

### 2.1. Ethics

The study was approved by the Bioethics Committee of Warsaw Management University. Participation in the study was voluntary; all respondents were informed that they could discontinue their participation at any time without any consequences. The study procedure was conducted in accordance with the Declaration of Helsinki. All participants received research information, fully understood the study’s purpose and gave signed informed consent to participate in the study.

### 2.2. Translation Procedure

The translation procedure was prepared to take into account previous translations of analysed scales and principles used in cross-cultural studies. In order to ensure adequate linguistic accuracy, as a first step, the original tools were translated from English into Polish by three researchers fluent in both Polish and English. The results of these three translations were analysed, and discrepancies were identified. The meanings of some terms were modified or changed to improve understandability and cultural appropriateness. In the next step, two other bilingual experts performed the back translation (from Polish to English). The back-translated scales were compared by a native English language speaker with the original versions to confirm the accuracy and relevance of the translation. Once minor discrepancies were corrected and agreed upon, the final version of the translation was checked by three experts in psychology. The prepared translation was used to carry out a pilot study to confirm the comprehensibility and accuracy of the sentences. The pilot study was carried out on a group of 10 volunteer nursing students. The students did not indicate that the presented sentences were incomprehensible or that they had problems answering any question. The scales prepared as described above were used to carry out the study.

### 2.3. Tools

#### 2.3.1. Metrics and Demographic Data

Before completing the psychological questionnaires, demographic data were collected from the individuals. These data included age, gender, height and body weight. On the basis of the declared height and weight data, the body mass index (BMI) was calculated.

#### 2.3.2. ATOP

According to the authors, the attitude toward obese people (ATOP) scale was built on the basis of the adapted attitude toward disabled persons scale. The ATOP scale was constructed from 20 items rated on a six-point scale. The respondents rated each item from −3 = strongly disagree to +3 = strongly agree [32]. The scale was constructed so that a higher score indicates a more positive attitude toward overweight and obese people.

The score calculation involved reversing the results from 13 negatively formulated items and adding the number 60 to the total score. Factor analysis, both the original version and the Turkish or Chinese versions, indicated a three-factor structure. The identified factors are different personality (ATOP–DP), social difficulties (ATOP–SD) and self-esteem (ATOP–SE). The different personality, as a factor, seems to reflect the attribution of negative or different personality traits, as well as inferior abilities to obese people. The social difficulties factor represents the perception of obese people as experiencing or creating social problems. The self-esteem factor refers to obese people’s perception and evaluation of themselves [32,33,34]. The above factors—different personality, social difficulties and self-esteem—are treated in most studies as a unidimensional scale [35,36]. Cronbach’s alpha coefficients range from 0.65 to 0.82 depending on the scale version [32,33,34].

#### 2.3.3. BAOP

The beliefs about obese persons (BAOP) scale was developed to represent the degree to which the respondent believes that obesity is controlled by an obese person. A higher scale score indicates a stronger belief that obese people cannot control their weight levels. The scale contains 8 items rated on a six-point scale from −3 = strongly disagree to +3 = strongly agree. In order to calculate the score, the 6 negatively formulated items are reversed and added to the score. A value of 24 is added to the final score [32]. The BAOP is a unidimensional scale which was confirmed in the original version and in the Turkish and Chinese versions. The internal consistency (Cronbach’s alpha) of the BAOP scale ranged from 0.64 to 0.82 [32,33,34].

#### 2.3.4. ORK–10

The obesity risk knowledge scale (ORK–10) was designed by Swift, Glazebrook and Macdonald. This 10-item test was designed to assess respondents’ knowledge of the health risks associated with obesity. The scale contains 10 questions, including true/false or I do not know answers. The respondent received 1 point for a correct answer; a wrong answer or an “I don’t know” answer is not scored. A maximum of 10 points can be collected. Higher scores indicated a better knowledge concerning obesity risks. The scale has a good internal consistency (Cronbach’s alpha = 0.8). In addition, statistically significant differences in mean scores were found between experts and non-experts, which confirmed the accuracy of the scale [37].

#### 2.3.5. Personality

The respondents’ personality traits were determined using the personality inventory NEO-FFI developed by Costa and McCrae [38]. This inventory, based on a five-factor model of personality, has been used for years in research, employee training and career counselling [39,40]. A shortened version of Costa and McCrae’s inventory contains 60 items (12 items per scale) and allows for a general diagnosis of the five basic personality traits.

### 2.4. Data Analysis

Data were analysed using SPSS software Version 27 (Warsaw, Poland) for Windows. Descriptive statistics and distribution tests were used to assess the distribution of variables. The performed analyses indicated that parametric tests could be used. In order to assess the adequacy of sampling, the Kaiser–Meyer–Olkin (KMO) test and Bartlett’s test of sphericity were applied. In order to examine the factor structure of the ATOP/BAOP scales, exploratory factor analysis was conducted using principal component analysis followed by a varimax rotation of the extracted factors. Internal consistency was tested using Cronbach’s alpha coefficients. *p*-values less than 0.05 were considered statistically significant.

## 3. Results

### 3.1. Study Sample Characteristics

The study was conducted on a group of 306 individuals, including 189 females (62%) and 117 males (38%). The mean BMI of the studied group was 23.2, SD = 3.7. The mean age of the subjects was 27.6, SD = 9.7, and ranged from 18 to 58 years. Due to the small number of individuals with a BMI below the normal range (N = 14), this group was not included in the comparative analyses. The other groups, i.e., with normal BMI (N = 214) and higher than normal BMI (N = 74) values, had sufficient numbers to perform the planned statistical analyses.

### 3.2. The Obesity Risk Knowledge 

Table 1 shows the original and translated questions comprising the ORK–10 scale. Reliability testing on the Polish sample (Cronbach’s alpha = 0.81) indicated good reliability similar to that of the original scale. Correlation analysis of ORK–10 scores with age and with BMI did not indicate the presence of statistically significant results. As ORK–10 is a test of knowledge and not a typical scale measuring a specific psychological variable, statistical psychometric procedures were not performed. These procedures are typical for scales that are required to achieve psychometric properties.

### 3.3. PL–ATOP Scale

#### 3.3.1. PL–ATOP Validity and Reliability

In order to determine whether the collected dataset was appropriate to perform a factorial analysis, the KMO sample adequacy test and Bartlett’s Sphericity Test were used. The KMO value was 0.73, indicating an adequate sample. The result for the Bartlett Sphericity Test showed statistical significance (χ^2^ = 1022.44, *p* < 0.001), which provided the basis for conducting a factorial analysis. These analyses confirmed the three-factor structure of the PL–ATOP, analogous to the original version. The three-factor structure accounted for 36.6% of the total variance. Factor 1, explaining 15.56% of the variance, consisted of eight items, which were marked as a different personality. Factor 2 comprised six items explaining 11.92% of the variance, which were described as social difficulties. The remaining six items in factor 3, explaining 9.13% of the variance, were marked as self-esteem. In order to assess the internal consistency of the scale, Cronbach’s alpha coefficients ranging from 0.61 to 0.68 for the subscales and 0.73 for the main scale score were calculated. The rotated factor solution and Cronbach’s alpha coefficient are presented in Table 2.

There were also separate analyses performed for the groups only according to the BMI criterion (normal and above normal). These analyses indicated that for the obese and overweight group, the total explained variance was higher and amounted to 43%.

#### 3.3.2. PL–ATOP Differences in Groups by Gender and BMI

A comparative analysis of the mean scores of the PL–ATOP scale and of the three separate subscales was made. The analysis showed that gender significantly differentiated mean scores on all scales (*p* < 0.01), with females obtaining higher scores than males. Cohen’s d analysis indicated a mean effect strength (PL–ATOP–0.62; PL–ATOP–DP–0.45; PL–ATOP–SD–0.50; PL–ATOP–SE–0.43). The analysis of differences, taking into account BMI (normal/above normal), indicated a statistically significant difference for the PL–ATOP–SE subscale (*p* = 0.01). However, this difference had a low strength effect (Cohen’s d = 0.35). Another analysis of differences in mean PL–ATOP values for the groups according to BMI criterion and gender as a sub-criterion was performed. The analysis showed significant differences in mean PL–ATOP–SE values (M = 8.05; SD = 3.72 for BMI: 18.5–25 vs. M = 4.40; SD = 3.10 for BMI > 25) and PL–ATOP–SD values (M = 12.00; SD = 3.83 for BMI: 18.8–25 vs. M = 14.20; SD = 5.33 for BMI > 25) only in males. In the case of females, BMI did not differentiate the mean scores of the PL–ATOP scale and its subscales. Figure 1 shows box plots of PL–ATOP scale scores for each group.

#### 3.3.3. Correlation and Regression Analysis of the PL–ATOP Scale

Table 3 shows the results of Pearson’s linear correlation analysis for PL–ATOP and subscales involving age, BMI, personality, ORK–10 and BAOP. A correlation analysis taking into account the criteria of gender and BMI (normal, above normal) was performed. A very strong correlation between the ATOP and the ORK–10 scores was observed. This correlation was high regardless of gender or body mass. A similar result was found for ATOP–DF and ATOP–SD subscales. Only the ATOP–SE scale correlated at a lower level with ORK–10. Much lower correlations were found when analysing personality. The correlation level ranged—depending on the personality trait—from no correlation to moderate correlation (r-Pearson = 0.52 for agreeableness and BMI > 25 group) and was dependent on gender and BMI. Linear regression analyses were also performed with PL–ATOP as the criterion variable. These analyses indicated that the ORK–10 score significantly explained more than 58% of the variation. Personality explained 20% of the variation with significant coefficients in the form of neuroticism, extraversion and agreeableness. An analogous regression analysis conducted only in a group of individuals with higher BMI showed that the ORK–10 score explained significantly over 62% of the variation. Personality traits were statistically insignificant.

### 3.4. BAOP Scale

#### 3.4.1. PL–BAOP Validity and Reliability

In order to determine whether the collected dataset was adequate to perform a factorial analysis, the KMO sample adequacy test and Bartlett’s Sphericity Test were conducted. The KMO score was 0.82, indicating sample adequacy. The result for the Bartlett Sphericity Test showed statistical significance (χ^2^ = 704.95, *p* < 0.000). It warranted a factorial analysis, and this analysis confirmed the univariate structure of the PL–BAOP, as in the original version of the tool. The factorial structure accounted for 42.61% of the total variance.

In order to assess the internal scale consistency, Cronbach’s alpha coefficient (with 0.76 value) was calculated. Item loadings and Cronbach’s alpha coefficient are shown in Table 4.

#### 3.4.2. PL–BAOP Differences in Groups by Gender and BMI

A comparative analysis of mean PL–BAOP scale scores was performed. It showed that gender significantly differentiated mean PL–BAOP scores (*p* < 0.01), with females obtaining higher scores than males. Cohen’s d analysis indicated a medium strength of the effect (Cohen’s d = 0.43). Further analysis of the differences in mean ATOP values for groups according to BMI by gender was performed. This analysis showed significant differences in mean PL–BAOP values only in males (M = 15.43; SD = 7.71 for BMI: 18.5–25 vs. M = 11.87; SD = 6.99 for BMI > 25). In females, BMI did not differentiate the mean BAOP scale scores.

Figure 2 shows box plots of PL–BAOP scale scores for each group.

#### 3.4.3. Correlation and Regression Analysis of the PL–BAOP Scale

Table 5 shows the results of Pearson’s linear correlation analysis for PL–BAOP with age, BMI, personality and PL–ATOP. Correlation analysis was performed for all subjects and grouped by criteria: gender and BMI (normal, above normal). The results indicated a very strong correlation between PL–BAOP and ORK–10 scores, and this correlation was strong regardless of gender or normal BMI. When analysing personality, a significant but low correlation was found only for agreeableness. Linear regression analyses were also performed with PL–BAOP as the criterion variable, which indicated that the ORK–10 score significantly explained 50% of the variation. Personality explained 3.7% of the variation with significant coefficients in agreeableness. An analogous regression analysis conducted only in the group with higher than normal BMI showed that the ORK–10 score explained more than 57% of the variation significantly. Personality, as in the PL–ATOP analysis, was found to be statistically insignificant.

## 4. Discussion

The main aim of our study was to present the validation of the Polish versions of two popular tools for measuring attitudes (PL–ATOP) and beliefs (PL–BAOP) to obesity conducted among people from the general (Caucasian) population, and a scale that diagnoses respondents’ knowledge of obesity-related health risks (PL–ORK–10).

PL–ATOP and PL–BAOP were translated and validated for the Polish population to be like the original versions of the scales. In both tools, the number of items was maintained because the factor load of each scale exceeded 0.40 (PL–BAOP-8 items, PL–ATOP-20 items). PL–ATOP and PL–BAOP showed good content relevance. The simplicity and reliability of the scales make both tools useful in research and clinical practice. There is also potential for their use in preventive and educational interventions with a wide range of audiences.

The attitudes towards obese people scale (PL–ATOP) has a three-factor structure as its original version. However, data indicating a different structure can be found in the literature, and the differences are usually explained by cultural differences and variations in the validation procedure and data analysis [33,41,42]. For both the total score and the three subscales, Cronbach’s α is close to the original values (in this study, it was 0.73 for the main score, 0.68 for the different personality scale, 0.64 for the social difficulties scale and 0.61 for the self-esteem scale). Factor I, which accounts for 15.56% of the variance, was named different personality because the items in it reflect negative or different characteristics attributed to obese people (e.g., personality traits). Factor II contains items concerned with causing social problems or events experienced by obese people. It accounted for 11.92% of the variance and was named social difficulties. Factor III-Self-Esteem accounted for 9.13% of the variance and contained items referring to how obese people evaluate themselves. It can therefore be concluded that the developed PL–ATOP scale has satisfactory psychometric properties to measure attitudes toward obese people.

Validation of the beliefs about obese persons scale (BAOP) revealed a one-factor structure accounting for 42.61% of the variance. The Cronbach’s α coefficient was 0.76 for the Polish version of the BAOP scale. Therefore, it can be assumed the scale can be used to measure beliefs about obese people.

The third tool used in the study, the PL–ORK–10 scale (obesity risk knowledge scale), showed good reliability as a measurement tool checking the level of respondents’ knowledge about obesity-related risks (Cronbach’s alpha coefficient was 0.81). Therefore, it can be used to assess the knowledge of the obesity health consequences for the adults in the Polish population.

In our study, a high correlation between the PL–ORK–10, PL–BAOP and PL–ATOP scales (0.75, *p* < 0.01; 0.77, *p* < 0.01) was observed, which confirmed our hypothesis assuming a positive relationship between knowledge of obesity health consequences and positive attitudes, as well as lack of prejudice to obese people. The obtained result indicated that the higher knowledge of the respondents about obesity dangers, the more positive their attitudes to obese people were and the stronger their conviction about the impossibility of body weight control was. A high level of knowledge about the medical consequences of obesity is associated with its causes being located more in independent factors rather than with self-control and with more positive attitudes to obese people. The high level of knowledge about obesity is not limited to knowing only the negative consequences of this disease but concerns a wide knowledge of the obesity phenomenon, taking into account its various causes and awareness of problems in obese people’s functioning. Recognition of obesity as a disease with its health consequences can also facilitate the perception of obesity as less dependent on the individual and foster more understanding and sympathetic attitudes. Understanding the situational complexity of obese people can reduce prejudice against overweight people and limit discrimination behaviour.

PL–ATOP scores were correlated with PL–BAOP scores (r-Pearson = 0.34). A similar moderate correlation was obtained between scales in other countries. There are more positive attitudes to obesity among those who believe that weight gain, to a large extent, is out of the obese person’s control than among those who believe that obesity can be controlled by the obese person. This finding is consistent with previous research using the cross-linguistic ATOP and BAOP scales [32,33,34]. 

Supporting these interpretations are also the findings between personality traits measured by the NEO-FFI and the validated scales. The results indicated that knowledge (in the case of obesity-related health risks), rather than personality, may determine attitudes toward overweight individuals. The study confirmed our hypothesis regarding the positive association of agreeableness with positive attitudes and lower levels of prejudice to obese people (PL–ATOP: r-Pearson = 0.38, *p* < 0.01; PL–BAOP: r-Pearson = 0.24, *p* < 0.001).

In order to better understand the relationships between the variables measured by PL–ATOP, PL–BAOP and PL–ORK–10, we also performed a series of analyses to determine whether gender and BMI differentiate the results obtained in our study. The themes of obesity, weight control pressures and prejudice against obese people are dependent on gender and excessive body weight. In our study, it was important to verify whether both of these variables would modify obtained results. In the presented material, ATOP and BAOP scores were modified by both gender and BMI of the participants. Females showed more positive attitudes toward overweight people and stronger beliefs regarding the inability of obese people to control their body mass.

Research on obesity and its associated social phenomena suggests that stereotypes underlie prejudice and discrimination against overweight people, consequently affecting their quality of life. Despite evidence of the partly automatic nature of stereotypes, under certain conditions, people are able to inhibit their influence on their own judgements. Research on ethnic stereotypes has shown that stereotype activation does not necessarily cause biased judgements and that reducing their use can foster anti-discriminatory behaviour [43]. This is consistent with evidence suggesting different motives that foster or limit the use of stereotypes [44,45,46,47]. In the view of the data, it is possible to conclude that interventions to reduce the stereotypes activation and their influence on evaluation formulation can support the reduction in negative evaluations towards obese people (also in terms of self-evaluation). It is also important to identify and reduce false and sometimes harmful beliefs that block or decrease the effectiveness of medical and psychotherapeutic interventions. Furthermore, knowledge as a factor associated with positive beliefs towards obesity also correlates positively in the group of overweight/obese people. Therefore, this factor can be used as a modifier of stereotypes and negative beliefs also among obese people so as to reduce the effects of self-stigmatisation. Overweight and obesity education among overweight and obese people can have positive results in terms of improving self-concepts, strengthening the individual’s resources, especially with persistence and effectiveness in maintaining a healthy lifestyle, and ultimately improving their overall wellbeing.

## 5. Conclusions

-Obesity risk knowledge predicts positive attitudes and beliefs toward obese people more than personality-The PL–ATOP and PL–BAOP are fully validated psychometric measurement tools;-The Polish version of the PL–ORK–10 is a fully useful tool that can also be successfully used with Polish-speaking respondents.

## 6. Limitation

The main limitation of the present study is the self-declared weight and height of the subjects. These data were used to calculate BMI. In our further studies, independent anthropometric measurements should be taken to eliminate this limitation.

## Figures and Tables

**Figure 1 ijerph-19-14977-f001:**
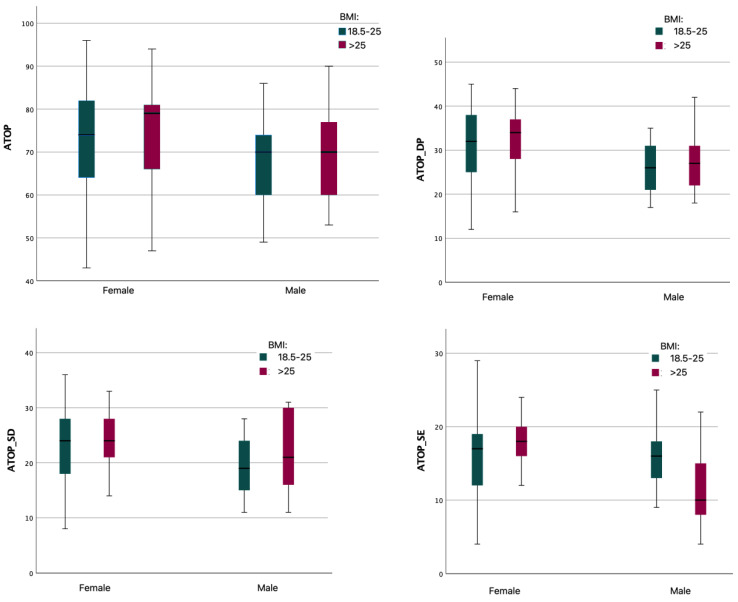
PL–ATOP results for gender and BMI groups.

**Figure 2 ijerph-19-14977-f002:**
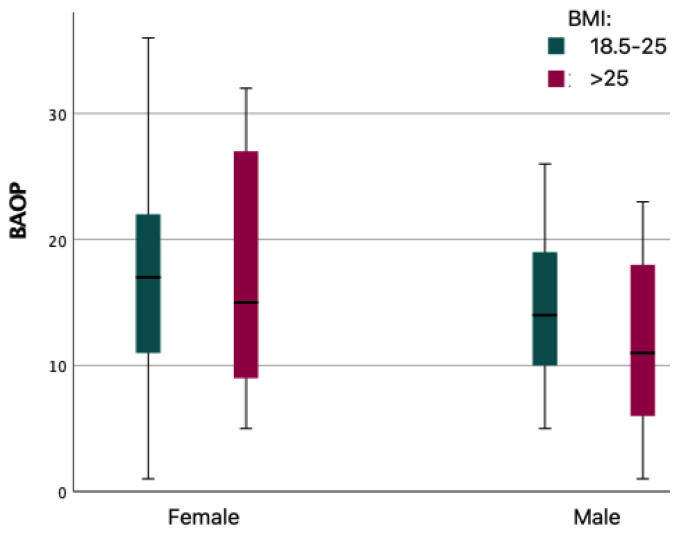
PL–BAOP for gender and BMI group.

**Table 1 ijerph-19-14977-t001:** Original and translated to polish items of ORK–10.

Items			True/False
1	EN	A person with a ‘beer-belly’ shaped stomach has an increased risk of getting diabetes	T
	PL	Osoba z brzuchem w kształcie “piwnego brzucha” ma zwiększone ryzyko zachorowania na cukrzycę	
2	EN	Obesity increases the risk of getting bowel cancer	T
	PL	Otyłość zwiększa ryzyko zachorowania na raka jelita grubego	
3	EN	An obese person who gets diabetes needs to lose at least 40% of their body weight for clear health benefits	F
	PL	Osoba otyła, która choruje na cukrzycę, musi stracić co najmniej 40% masy ciała, aby uzyskać wyraźne korzyści zdrowotne	
4	EN	Obese people can expect to live as long as nonobese people	F
	PL	Osoby otyłe mogą spodziewać się, że będą żyły tak długo jak osoby nieotyłe	
5	EN	Obesity increases the risk of getting breast cancer after the menopause	T
	PL	Otyłość zwiększa ryzyko zachorowania na raka piersi po menopauzie	
6	EN	Obesity is more of a risk to health for people from South Asia (e.g., India and Pakistan) than it is for White Europeans	T
	PL	Otyłość jest większym zagrożeniem dla zdrowia dla osób z Azji Południowej (np. Indii i Pakistanu) niż dla białych Europejczyków	
7	EN	There is no major health benefit if an obese person who gets diabetes, loses weight	F
	PL	Nie stanowi istotnych korzyści zdrowotnych, jeśli osoba otyła, która zachoruje na cukrzycę, straci na wadze	
8	EN	Obesity does not increase the risk of developing high blood pressure	F
	PL	Otyłość nie zwiększa ryzyka rozwoju wysokiego ciśnienia krwi	
9	EN	It is better for a person’s health to have fat around the hips and thighs than around the stomach and waist	T
	PL	Lepiej dla zdrowia człowieka jest mieć tłuszcz wokół bioder i ud niż wokół brzucha i talii	
10	EN	Obesity increases the risk of getting a food allergy	F
	PL	Otyłość zwiększa ryzyko zachorowania na alergię pokarmową	

**Table 2 ijerph-19-14977-t002:** PL–ATOP Factor Analysis.

Factors	ATOP Items			Factor Loading	% of the Variance (Cronbach’s Alpha)	
Different personality	4	EN	Obese workers cannot be as successful as other workers.	0.53	15.56% (0.68)	36.62% (0.73)
		PL	Otyli pracownicy nie mogą odnosić takich sukcesów jak inni pracownicy.			
	6	EN	Severely obese people are usually untidy	0.54		
		PL	Poważnie otyli ludzie są zwykle niechlujni.			
	11	EN	Obese people are often less aggressive than nonobese people.	0.45		
		PL	Osoby otyłe są często mniej agresywne niż osoby nieotyłe.			
	12	EN	Most obese people have different personalities than nonobese people	0.59		
		PL	Większość osób otyłych ma inną osobowość niż osoby nieotyłe.			
	14	EN	Most obese people resent normal weight people.	0.65		
		PL	Większość osób otyłych nie lubi ludzi o prawidłowej masie ciała.			
	15	EN	Obese people are more emotional than nonobese people.	0.61		
		PL	Osoby otyłe są bardziej emocjonalne niż osoby nieotyłe.			
	17	EN	Obese people are just as healthy as nonobese people	0.63		
		PL	Osoby otyłe są tak samo zdrowe jak osoby nieotyłe.			
	20	EN	One of the worst things that could happen to a person would be for him to become obese.	0.64		
		PL	Jedną z najgorszych rzeczy, jaka może się przydarzyć w życiu jest otyłość.			
Social difficulties	5	EN	Most nonobese people would not want to marry anyone who is obese.	0.72	11.92% (0.64)	
		PL	Większość osób nieotyłych nie chciałaby poślubić nikogo, kto jest otyły.			
	7	EN	Obese people are usually sociable.	0.40		
		PL	Osoby otyłe są zazwyczaj towarzyskie.			
	10	EN	Most people feel uncomfortable when they associate with obese people.	0.62		
		PL	Większość ludzi czuje się nieswojo, gdy ma kontakt z osobami otyłymi.			
	16	EN	Obese people should not expect to lead normal lives.	0.41		
		PL	Osoby otyłe nie powinny oczekiwać, że będą prowadziły normalne życie.			
	18	EN	Obese people are just as sexually attractive as nonobese people.	0.63		
		PL	Osoby otyłe są tak samo atrakcyjne seksualnie jak osoby nieotyłe.			
	19	EN	Obese people tend to have family problems.	0.49		
		PL	Osoby otyłe mają zwykle problemy rodzinne.			
Self-esteem	1	EN	Obese people are as happy as nonobese people.	0.70	9.13% (0.61)	
		PL	Osoby otyłe są tak samo szczęśliwe jak osoby nieotyłe.			
	2	EN	Most obese people feel that they are not as good as other people.	0.42		
		PL	Większość osób otyłych uważa, że nie są oni tak dobrzy jak inni.			
	3	EN	Most obese people are more self-conscious than other people.	0.48		
		PL	Większość osób otyłych jest bardziej świadomymi siebie niż inne osoby.			
	8	EN	Most obese people are not dissatisfied with themselves.	0.71		
		PL	Większość osób otyłych nie jest z siebie niezadowolona.			
	9	EN	Obese people are just as self-confident as other people.	0.76		
		PL	Osoby otyłe są tak samo pewne siebie jak inne osoby.			
	13	EN	Very few obese people are ashamed of their weight.	0.43		
		PL	Bardzo niewiele osób otyłych wstydzi się swojej wagi.			

**Table 3 ijerph-19-14977-t003:** Pearson’s linear correlation analysis for PL–ATOP.

	Variables	GENDER		BMI		ALL
		Female	Male	<25	>25	
PL–ATOP	Age	−0.05	0.33 **	0.01	0.265 **	0.06
	BMI	−0.06	0.02	−0.01	−0.33 **	−0.11
	Neuroticism	−0.07	−0.18	0.05	0.15	0.06
	Extraversion	0.27 **	0.22	0.032 **	−0.11	0.21 *
	Openness	0.15	0.14	0.13	0.01	0.06
	Agreeableness	0.36 **	0.27	0.44 **	0.38	0.38 **
	Conscientiousness	0.09	0.06	0.19	−0.16	0.10
	Obesity Risk Knowledge	0.75 **	0.81 **	0.75 **	0.79 **	0.77 **
	BAOP	0.27 **	0.55 **	0.35 **	0.45 **	0.34 *
PL–ATOP-Different Personality	Age	−0.01	0.49 **	0.09	0.19	0.11
	BMI	−0.07	0.21	−0.03	−0.31 **	−0.05
	Neuroticism	−0.12	−0.07	0.01	0.31	0.03
	Extraversion	0.37 **	0.20	0.38 **	−0.42 *	0.27 *
	Openness	0.14	0.02	0.15	−0.36	0.03
	Agreeableness	0.29 **	0.20	0.38 **	0.24	0.31 **
	Conscientiousness	0.08	−0.09	0.13	−0.30	0.05
	Obesity Risk Knowledge	0.72 **	0.73 **	0.74 **	0.69 **	0.72 **
	BAOP	0.26 **	0.29 **	0.35 **	0.25 *	0.32 *
PL–ATOP-Social Difficulties	Age	−0.07	0.13	−0.01	0.08	−0.01
	BMI	−0.04	0.11	0.08	−0.31 **	−0.06
	Neuroticism	0.13	−0.22	0.17	0.02	0.16
	Extraversion	−0.05	0.04	0.04	−0.02	−0.04
	Openness	0.18	0.45 *	0.17	0.42 **	0.20 *
	Agreeableness	0.25 *	0.35 *	0.35 **	0.52 *	0.33 *
	Conscientiousness	0.03	0.19	0.09	−0.02	0.09
	Obesity Risk Knowledge	0.70 **	0.54 **	0.65 **	0.74 **	0.68 **
	BAOP	0.28 **	0.34 **	0.29 **	0.44 **	0.29 *
PL–ATOP-Self-Esteem	Age	−0.03	−0.08	−0.1	0.33 **	−0.01
	BMI	−0.02	−0.39 **	−0.08	−0.04	−0.16 *
	Neuroticism	−0.16	−0.04	−0.08	−0.03	−0.02
	Extraversion	0.27 **	0.18	0.29 **	0.22	0.21 *
	Openness	0.01	−0.24	−0.07	−0.17	−0.13
	Agreeableness	0.32 **	−0.01	0.30 **	−0.07	0.21 *
	Conscientiousness	0.13	0.06	0.26 **	−0.02	0.116
	Obesity Risk Knowledge	0.16	0.59 **	0.27 **	0.19	0.24 **
	BAOP	0.04	0.50 **	0.13	0.25 *	0.12 *

* *p* < 0.05; ** *p* < 0.01.

**Table 4 ijerph-19-14977-t004:** PL–BAOP Factor Analysis.

BAOP Items			Factor Loading	% of the Variance (Cronbach’s Alpha)
1	EN	Obesity often occurs when eating is used as a form of compensation for lack of love or attention.	0.59	42.61% (0.76)
	PL	Otyłość często pojawia się, gdy jedzenie jest wykorzystywane jako forma rekompensaty za brak miłości lub uwagi.		
2	EN	In many cases, obesity is the result of a biological disorder.	0.40	
	PL	W wielu przypadkach otyłość jest wynikiem zaburzeń o podłożu medycznym.		
3	EN	Obesity is usually caused by overeating.	0.76	
	PL	Otyłość jest zwykle spowodowana przejadaniem się.		
4	EN	Most obese people cause their problem by not getting enough exercise.	0.82	
	PL	U większości osób otyłych problem z otyłością jest powodowany przez brak wystarczającej ilości ruchu.		
5	EN	Most obese people eat more than nonobese people.	0.80	
	PL	Większość osób otyłych je więcej niż osoby zdrowe.		
6	EN	The majority of obese people have poor eating habits that lead to their obesity.	0.81	
	PL	Większość osób otyłych ma złe nawyki żywieniowe, które prowadzą do otyłości.		
7	EN	Obesity is rarely caused by a lack of willpower.	0.60	
	PL	Otyłość rzadko kiedy jest spowodowana brakiem siły woli.		
8	EN	People can be addicted to food, just as others are addicted to drugs, and these people usually become obese.	0.58	
	PL	Ludzie mogą być uzależnieni od jedzenia, tak jak inni są uzależnieni od narkotyków. Takie uzależnienie jest zwykle przyczyną otyłości.		

**Table 5 ijerph-19-14977-t005:** Pearson’s linear correlation analysis for PL–BAOP.

Variables	GENDER		BMI		ALL Individuals
	Female	Male	<25	>25	
Age	0.01	0.08	−0.07	0.38 **	0.05
BMI	0.09	−0.13	0.11	0.03	−0.03
Neuroticism	0.01	0.10	0.09	0.38	0.14
Extraversion	0.07	−0.04	0.07	0.01	0.01
Openness	0.07	−0.13	0.01	−0.09	−0.06
Agreeableness	0.22 *	0.13	0.22	0.29	0.24 **
Conscientiousness	0.01	−0.02	0.11	−0.08	0.02
Obesity Risk Knowledge	0.73 **	0.83 **	0.74 **	0.76 **	0.75 **
ATOP	0.27 **	0.55 **	0.35 **	0.45 **	0.34 **
ATOP–DP	0.26 **	0.29 **	0.35 **	0.25 *	0.32 **
ATOP–SD	0.28 ***	0.34 **	0.29 **	0.44 **	0.29 **
ATOP–SE	0.04	0.50 *	0.13	0.25 *	0.12 *

* *p* < 0.05; ** *p* < 0.01.

## Data Availability

Not applicable.

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
