# Peer review of "Reliable Knowledge about Obesity Risk, Rather Than Personality, Is Associated with Positive Beliefs towards Obese People: Investigating Attitudes and Beliefs about Obesity, and Validating the Polish Versions of ATOP, BAOP and ORK–10 Scales"

_ijerph, 2022, doi:10.3390/ijerph192214977_

Round 1
Reviewer 1 Report
The study is well conducted. The Methodology is appropriated and the results are clear.
The introduction is very redundant. I suggest to authors the reduction of the first part of the introduction.
Author Response
Dear Reviewer, thank you for your comments. All your suggestions have been taken into account.
The reviewer suggests shortening the introduction in the submitted material.
The article covers a broad spectrum of issues and, in our view, the introduction should accurately describe the issues in the field. However, given the suggestion, the manuscript has been revised and the introduction has been reduced.
Reviewer 2 Report
The authors did not provide any data in their results section within the abstract to support their conclusion 'Regression analysis indicated that knowledge of obesity risk predicted positive attitudes towards obese people by more 25 than 50%, in turn personality only 20%'. Conclusions should be based off of results presented so this conclusion within the abstract section came off as out of the blue if the abstract is reviewed as a stand-alone document. However, within the main text, since the results of the regression analysis was presented, it flowed sequentially to have that conclusion at the end.
Author Response
Dear Reviewer, thank you for your comments. All your suggestions have been taken into account.
The reviewer in the submitted material suggests a change in the Abstract in order to present more clearly the results that formed the basis for the main conclusion.
This suggestion is fully justified. The Abstract has been restructured. Some of the data were moved to the results section and supplemented. The conclusions section has also been completed.